# Optimal Linear Estimation under Unknown Nonlinear Transform

**Xinyang Yi**
The University of Texas at Austin
yixy@utexas.edu

**Zhaoran Wang**
Princeton University
zhaoran@princeton.edu

**Constantine Caramanis**
The University of Texas at Austin
constantine@utexas.edu

**Han Liu**
Princeton University
hanliu@princeton.edu

## Abstract

Linear regression studies the problem of estimating a model parameter $\boldsymbol{\beta}^* \in \mathbb{R}^p$, from $n$ observations $\{(y_i, \mathbf{x}_i)\}_{i=1}^n$ from linear model $y_i = \langle \mathbf{x}_i, \boldsymbol{\beta}^* \rangle + \epsilon_i$. We consider a significant generalization in which the relationship between $\langle \mathbf{x}_i, \boldsymbol{\beta}^* \rangle$ and $y_i$ is noisy, quantized to a single bit, potentially nonlinear, noninvertible, as well as unknown. This model is known as the single-index model in statistics, and, among other things, it represents a significant generalization of one-bit compressed sensing. We propose a novel spectral-based estimation procedure and show that we can recover $\boldsymbol{\beta}^*$ in settings (i.e., classes of link function $f$) where previous algorithms fail. In general, our algorithm requires only very mild restrictions on the (unknown) functional relationship between $y_i$ and $\langle \mathbf{x}_i, \boldsymbol{\beta}^* \rangle$. We also consider the high dimensional setting where $\boldsymbol{\beta}^*$ is sparse, and introduce a two-stage nonconvex framework that addresses estimation challenges in high dimensional regimes where $p \gg n$. For a broad class of link functions between $\langle \mathbf{x}_i, \boldsymbol{\beta}^* \rangle$ and $y_i$, we establish minimax lower bounds that demonstrate the optimality of our estimators in both the classical and high dimensional regimes.

## 1 Introduction

We consider a generalization of the one-bit quantized regression problem, where we seek to recover the regression coefficient $\boldsymbol{\beta}^* \in \mathbb{R}^p$ from one-bit measurements. Specifically, suppose that $\boldsymbol{X}$ is a random vector in $\mathbb{R}^p$ and $Y$ is a binary random variable taking values in $\{-1, 1\}$. We assume the conditional distribution of $Y$ given $\boldsymbol{X}$ takes the form

$$\mathbb{P}(Y = 1 | \boldsymbol{X} = \boldsymbol{x}) = \frac{1}{2} f(\langle \boldsymbol{x}, \boldsymbol{\beta}^* \rangle) + \frac{1}{2}, \tag{1.1}$$

where $f : \mathbb{R} \to [-1, 1]$ is called the *link function*. We aim to estimate $\boldsymbol{\beta}^*$ from $n$ i.i.d. observations $\{(y_i, \mathbf{x}_i)\}_{i=1}^n$ of the pair $(Y, \boldsymbol{X})$. In particular, we assume the link function $f$ is unknown. Without any loss of generality, we take $\boldsymbol{\beta}^*$ to be on the unit sphere $\mathbb{S}^{p-1}$ since its magnitude can always be incorporated into the link function $f$.

The model in (1.1) is simple but general. Under specific choices of the link function $f$, (1.1) immediately leads to many practical models in machine learning and signal processing, including logistic regression and one-bit compressed sensing. In the settings where the link function is assumed to be known, a popular estimation procedure is to calculate an estimator that minimizes a certain loss

function. However, for particular link functions, this approach involves minimizing a nonconvex objective function for which the global minimizer is in general intractable to obtain. Furthermore, it is difficult or even impossible to know the link function in practice, and a poor choice of link function may result in inaccurate parameter estimation and high prediction error. We take a more general approach, and in particular, target the setting where $f$ is unknown. We propose an algorithm that can estimate the parameter $\boldsymbol{\beta}^*$ in the absence of prior knowledge on the link function $f$. As our results make precise, our algorithm succeeds as long as the function $f$ satisfies a single moment condition. As we demonstrate, this moment condition is only a mild restriction on $f$. In particular, our methods and theory are widely applicable even to the settings where $f$ is non-smooth, e.g., $f(z) = \text{sign}(z)$, or noninvertible, e.g., $f(z) = \sin(z)$.

In particular, as we show in §2, our restrictions on $f$ are sufficiently flexible so that our results provide a unified framework that encompasses a broad range of problems, including logistic regression, one-bit compressed sensing, one-bit phase retrieval as well as their robust extensions. We use these important examples to illustrate our results, and discuss them at several points throughout the paper.

**Main contributions.** The key conceptual contribution of this work is a novel use of the method of moments. Rather than considering moments of the covariate, $\boldsymbol{X}$, and the response variable, $Y$, we look at moments of differences of covariates, and differences of response variables. Such a simple yet critical observation enables everything that follows and leads to our spectral-based procedure.

We also make two theoretical contributions. First, we simultaneously establish the statistical and computational rates of convergence of the proposed spectral algorithm. We consider both the *low dimensional setting* where the number of samples exceeds the dimension and *the high dimensional setting* where the dimensionality may (greatly) exceed the number of samples. In both these settings, our proposed algorithm achieves the same statistical rate of convergence as that of linear regression applied on data generated by the linear model without quantization. Second, we provide minimax lower bounds for the statistical rate of convergence, and thereby establish the optimality of our procedure within a broad model class. In the low dimensional setting, our results obtain the optimal rate with the optimal sample complexity. In the high dimensional setting, our algorithm requires estimating a sparse eigenvector, and thus our sample complexity coincides with what is believed to be the best achievable via polynomial time methods [2]; the error rate itself, however, is information-theoretically optimal. We discuss this further in §3.4.

**Related works.** Our model in (1.1) is close to the *single-index model* (SIM) in statistics. In the SIM, we assume that the response-covariate pair $(Y, \boldsymbol{X})$ is determined by

$$Y = f(\langle \boldsymbol{X}, \boldsymbol{\beta}^* \rangle) + W \tag{1.2}$$

with unknown link function $f$ and noise $W$. Our setting is a special case of this, as we restrict $Y$ to be a binary random variable. The single index model is a classical topic, and therefore there is extensive literature – too much to exhaustively review it. We therefore outline the pieces of work most relevant to our setting and our results. For estimating $\boldsymbol{\beta}^*$ in (1.2), a feasible approach is $M$-estimation [8, 9, 12], in which the unknown link function $f$ is jointly estimated using nonparametric estimators. Although these $M$-estimators have been shown to be consistent, they are not computationally efficient since they involve solving a nonconvex optimization problem. Another approach to estimate $\boldsymbol{\beta}^*$ is named the *average derivative estimator* (ADE; [24]). Further improvements of ADE are considered in [13, 22]. ADE and its related methods require that the link function $f$ is at least differentiable, and thus excludes important models such as one-bit compressed sensing with $f(z) = \text{sign}(z)$. Beyond estimating $\boldsymbol{\beta}^*$, the works in [15, 16] focus on iteratively estimating a function $f$ and vector $\boldsymbol{\beta}$ that are good for prediction, and they attempt to control the generalization error. Their algorithms are based on isotonic regression, and are therefore only applicable when the link function is monotonic and satisfies Lipschitz constraints. The work discussed above focuses on the low dimensional setting where $p \ll n$. Another related line of works is *sufficient dimension reduction*, where the goal is to find a subspace $\mathbf{U}$ of the input space such that the response $Y$ only depends on the projection $\mathbf{U}^\top \boldsymbol{X}$. Single-index model and our problem can be regarded as special cases of this problem as we are primarily interested in recovering a one-dimensional subspace. Due to space limit, we refer readers to the long version of this paper for a detailed survey [29].

In the high dimensional regime with $p \gg n$ and $\boldsymbol{\beta}^*$ has some structure (for us this means sparsity), we note there exists some recent progress [1] on estimating $f$ via PAC Bayesian methods. In the special case when $f$ is linear function, sparse linear regression has attracted extensive study over the years. The recent work by Plan et al. [21] is closest to our setting. They consider the setting of normal covariates, $\boldsymbol{X} \sim \mathcal{N}(\boldsymbol{0}, \mathbf{I}_p)$, and they propose a marginal regression estimator for estimating $\boldsymbol{\beta}^*$, that, like our approach, requires no prior knowledge about $f$. Their proposed algorithm relies on the assumption that $\mathbb{E}_{z \sim \mathcal{N}(0,1)}[zf(z)] \neq 0$, and hence cannot work for link functions that are even. As we will describe below, our algorithm is based on a novel moment-based estimator, and avoids requiring such a condition, thus allowing us to handle even link functions under a very mild moment restriction, which we describe in detail below. Generally, the work in [21] requires different conditions, and thus beyond the discussion above, is not directly comparable to the work here. In cases where both approaches apply, the results are minimax optimal.

## 2 Example models

In this section, we discuss several popular (and important) models in machine learning and signal processing that fall into our general model (1.1) under specific link functions. Variants of these models have been studied extensively in the recent literature. These examples trace through the paper, and we use them to illustrate the details of our algorithms and results.

**Logistic regression.** In logistic regression (LR), we assume that $\mathbb{P}(Y = 1|\boldsymbol{X} = \boldsymbol{x}) = \frac{1}{1+\exp(-\langle\boldsymbol{x},\boldsymbol{\beta}^*\rangle-\zeta)}$, where $\zeta$ is the intercept. The link function corresponds to $f(z) = \frac{\exp(z+\zeta)-1}{\exp(z+\zeta)+1}$. One robust variant of LR is called *flipped logistic regression*, where we assume that the labels $Y$ generated from standard LR model are flipped with probability $p_{\mathrm{e}}$, i.e., $\mathbb{P}(Y = 1|\boldsymbol{X} = \boldsymbol{x}) = \frac{1-p_{\mathrm{e}}}{1+\exp(-\langle\boldsymbol{x},\boldsymbol{\beta}^*\rangle-\zeta)} + \frac{p_{\mathrm{e}}}{1+\exp(\langle\boldsymbol{x},\boldsymbol{\beta}^*\rangle+\zeta)}$. This reduces to the standard LR model when $p_{\mathrm{e}} = 0$. For flipped LR, the link function $f$ can be written as

$$f(z) = \frac{\exp(z+\zeta)-1}{\exp(z+\zeta)+1} + 2p_{\mathrm{e}} \cdot \frac{1-\exp(z+\zeta)}{1+\exp(z+\zeta)}. \tag{2.1}$$

Flipped LR has been studied by [19, 25]. In both papers, estimating $\boldsymbol{\beta}^*$ is based on minimizing some surrogate loss function involving a certain tuning parameter connected to $p_{\mathrm{e}}$. However, $p_{\mathrm{e}}$ is unknown in practice. In contrast to their approaches, our method does not hinge on the unknown parameter $p_{\mathrm{e}}$. Our approach has the same formulation for both standard and flipped LR, thus unifies the two models.

**One-bit compressed sensing.** One-bit compressed sensing (CS) aims at recovering sparse signals from quantized linear measurements (see e.g., [11, 20]). In detail, we define $\mathbb{B}_0(s,p) := \{\boldsymbol{\beta} \in \mathbb{R}^p : |\operatorname{supp}(\boldsymbol{\beta})| \leq s\}$ as the set of sparse vectors in $\mathbb{R}^p$ with at most $s$ nonzero elements. We assume $(Y, \boldsymbol{X}) \in \{-1, 1\} \times \mathbb{R}^p$ satisfies

$$Y = \operatorname{sign}(\langle\boldsymbol{X}, \boldsymbol{\beta}^*\rangle), \tag{2.2}$$

where $\boldsymbol{\beta}^* \in \mathbb{B}_0(s,p)$. In this paper, we also consider its robust version with noise $\epsilon$, i.e., $Y = \operatorname{sign}(\langle\boldsymbol{X}, \boldsymbol{\beta}^*\rangle + \epsilon)$. Assuming $\epsilon \sim \mathcal{N}(0, \sigma^2)$, the link function $f$ of robust 1-bit CS thus corresponds to

$$f(z) = 2\int_0^\infty \frac{1}{\sqrt{2\pi}\sigma} e^{-(u-z)^2/2\sigma^2} \mathrm{d}u - 1. \tag{2.3}$$

Note that (2.2) also corresponds to the probit regression model without the sparse constraint on $\boldsymbol{\beta}^*$. Throughout the paper, we do not distinguish between the two model names. Model (2.2) is referred to as one-bit compressed sensing even in the case where $\boldsymbol{\beta}^*$ is not sparse.

**One-bit phase retrieval**. The goal of phase retrieval (e.g., [5]) is to recover signals based on linear measurements with phase information erased, i.e., pair $(Y, \boldsymbol{X}) \in \mathbb{R} \times \mathbb{R}^p$ is determined by equation $Y = |\langle\boldsymbol{X}, \boldsymbol{\beta}^*\rangle|$. Analogous to one-bit compressed sensing, we consider a new model named *one-bit phase retrieval* where the linear measurement with phase information erased is quantized to one bit. In detail, pair $(Y, \boldsymbol{X}) \in \{-1, 1\} \times \mathbb{R}^p$ is linked through $Y = \operatorname{sign}(|\langle\boldsymbol{X}, \boldsymbol{\beta}^*\rangle| - \theta)$, where $\theta$ is the quantization threshold. Compared with one-bit compressed sensing, this problem is more difficult because $Y$ only depends on $\boldsymbol{\beta}^*$ through the magnitude of $\langle\boldsymbol{X}, \boldsymbol{\beta}^*\rangle$ instead of the value of $\langle\boldsymbol{X}, \boldsymbol{\beta}^*\rangle$. Also, it is more difficult than the original phase retrieval problem due to the additional quantization.

Using our general model, The link function thus corresponds to

$$f(z) = \text{sign}(|z| - \theta). \tag{2.4}$$

It is worth noting that, unlike previous models, here $f$ is neither odd nor monotonic.

## 3 Main results

We now turn to our algorithms for estimating $\boldsymbol{\beta}^*$ in both low and high dimensional settings. We first introduce a second moment estimator based on pairwise differences. We prove that the eigenstructure of the constructed second moment estimator encodes the information of $\boldsymbol{\beta}^*$. We then propose algorithms to estimate $\boldsymbol{\beta}^*$ based upon this second moment estimator. In the high dimensional setting where $\boldsymbol{\beta}^*$ is sparse, computing the top eigenvector of our pairwise-difference matrix reduces to computing a sparse eigenvector. Beyond algorithms, we discuss minimax lower bound in §3.5. We present simulation results in §3.6

### 3.1 Conditions for success

We now introduce several key quantities, which allow us to state precisely the conditions required for the success of our algorithm.

**Definition 3.1.** *For any (unknown) link function, $f$, define the quantity $\phi(f)$ as follows:*

$$\phi(f) := \mu_1^2 - \mu_0\mu_2 + \mu_0^2. \tag{3.1}$$

*where $\mu_0$, $\mu_1$ and $\mu_2$ are given by*

$$\mu_k := \mathbb{E}\big[f(Z)Z^k\big], \quad k = 0, 1, 2\ldots, \tag{3.2}$$

*where $Z \sim \mathcal{N}(0,1)$.*

As we discuss in detail below, the key condition for success of our algorithm is $\phi(f) \neq 0$. As we show below, this is a relatively mild condition, and in particular, it is satisfied by the three examples introduced in §2. For odd and monotonic $f$, $\phi(f) > 0$ unless $f(z) = 0$ for all $z$ in which case no algorithm is able to recover $\boldsymbol{\beta}^*$. For even $f$, we have $\mu_1 = 0$. Thus $\phi(f) \neq 0$ if and only if $\mu_0 \neq \mu_2$.

### 3.2 Second moment estimator

We describe a novel moment estimator that enables our algorithm. Let $\{(y_i, \mathbf{x}_i)\}_{i=1}^n$ be the $n$ i.i.d. observations of $(Y, \boldsymbol{X})$. Assuming without loss of generality that $n$ is even, we consider the following key transformation

$$\Delta y_i := y_{2i} - y_{2i-1}, \quad \Delta \mathbf{x}_i := \mathbf{x}_{2i} - \mathbf{x}_{2i-1}, \tag{3.3}$$

for $i = 1, 2, ..., n/2$. Our procedure is based on the following second moment

$$\mathbf{M} := \frac{2}{n} \sum_{i=1}^{n/2} \Delta y_i^2 \Delta \mathbf{x}_i \Delta \mathbf{x}_i^\top \in \mathbb{R}^{p \times p}. \tag{3.4}$$

The intuition behind this second moment is as follows. By (1.1), the variation of $\boldsymbol{X}$ along the direction $\boldsymbol{\beta}^*$ has the largest impact on the variation of $\langle \boldsymbol{X}, \boldsymbol{\beta}^* \rangle$. Thus, the variation of $Y$ directly depends on the variation of $\boldsymbol{X}$ along $\boldsymbol{\beta}^*$. Consequently, $\{(\Delta y_i, \Delta \mathbf{x}_i)\}_{i=1}^{n/2}$ encodes the information of such a dependency relationship. In the following, we make this intuition more rigorous by analyzing the eigenstructure of $\mathbb{E}(\mathbf{M})$ and its relationship with $\boldsymbol{\beta}^*$.

**Lemma 3.2.** *For $\boldsymbol{\beta}^* \in \mathbb{S}^{p-1}$, we assume that $(Y, \boldsymbol{X}) \in \{-1, 1\} \times \mathbb{R}^p$ satisfies (1.1). For $\boldsymbol{X} \sim \mathcal{N}(\mathbf{0}, \mathbf{I}_p)$, we have*

$$\mathbb{E}(\mathbf{M}) = 4\phi(f) \cdot \boldsymbol{\beta}^* \boldsymbol{\beta}^{*\top} + 4(1 - \mu_0^2) \cdot \mathbf{I}_p, \tag{3.5}$$

*where $\mu_0$ and $\phi(f)$ are defined in (3.2) and (3.1).*

Lemma 3.2 proves that $\boldsymbol{\beta}^*$ is the leading eigenvector of $\mathbb{E}(\mathbf{M})$ as long as the eigengap $\phi(f)$ is positive. If instead we have $\phi(f) < 0$, we can use a related moment estimator which has analogous properties.

To this end, define $\mathbf{M}' := \frac{2}{n}\sum_{i=1}^{n/2}(y_{2i} + y_{2i-1})^2 \Delta\mathbf{x}_i \Delta\mathbf{x}_i^\top$. In parallel to Lemma 3.2, we have a similar result for $\mathbf{M}'$ as stated below.

**Corollary 3.3.** *Under the setting of Lemma 3.2,*

$$\mathbb{E}(\mathbf{M}') = -4\phi(f) \cdot \boldsymbol{\beta}^* \boldsymbol{\beta}^{*\top} + 4(1 + \mu_0^2) \cdot \mathbf{I}_p.$$

Corollary 3.3 therefore shows that when $\phi(f) < 0$, we can construct another second moment estimator $\mathbf{M}'$ such that $\boldsymbol{\beta}^*$ is the leading eigenvector of $\mathbb{E}(\mathbf{M}')$. As discussed above, this is precisely the setting for one-bit phase retrieval when the quantization threshold in (3.1) satisfies $\theta < \theta_{\mathrm{m}}$. For simplicity of the discussion, hereafter we assume that $\phi(f) > 0$ and focus on the second moment estimator $\mathbf{M}$ defined in (3.4).

A natural question to ask is whether $\phi(f) \neq 0$ holds for specific models. The following lemma demonstrates exactly this, for the example models introduced in §2.

**Lemma 3.4.** (a) *Consider the flipped logistic regression where $f$ is given in (2.1). By setting the intercept to be $\zeta = 0$, we have $\phi(f) \gtrsim (1 - 2p_{\mathrm{e}})^2$. (b) For robust one-bit compressed sensing where $f$ is given in (2.3). We have $\phi(f) \gtrsim \min\left\{\left(\frac{1-\sigma^2}{1+\sigma^2}\right)^2, \frac{C'\sigma^4}{(1+\sigma^3)^2}\right\}$. (c) For one-bit phase retrieval where $f$ is given in (2.4). For $Z \sim \mathcal{N}(0,1)$, we let $\theta_{\mathrm{m}}$ be the median of $|Z|$, i.e., $\mathbb{P}(|Z| \geq \theta_{\mathrm{m}}) = 1/2$. We have $|\phi(f)| \gtrsim \theta|\theta - \theta_{\mathrm{m}}|\exp(-\theta^2)$ and $\mathrm{sign}[\phi(f)] = \mathrm{sign}(\theta - \theta_{\mathrm{m}})$. We thus obtain $\phi(f) > 0$ for $\theta > \theta_{\mathrm{m}}$.*

### 3.3 Low dimensional recovery

We consider estimating $\boldsymbol{\beta}^*$ in the classical (low dimensional) setting where $p \ll n$. Based on the second moment estimator $\mathbf{M}$ defined in (3.4), estimating $\boldsymbol{\beta}^*$ amounts to solving a noisy eigenvalue problem. We solve this by a simple iterative algorithm: provided an initial vector $\boldsymbol{\beta}^0 \in \mathbb{S}^{p-1}$ (which may be chosen at random) we perform power iterations as shown in Algorithm 1.

**Theorem 3.5.** *We assume $\boldsymbol{X} \sim \mathcal{N}(\mathbf{0}, \mathbf{I}_p)$ and $(Y, \boldsymbol{X})$ follows (1.1). Let $\{(y_i, \mathbf{x}_i)\}_{i=1}^n$ be $n$ i.i.d. samples of response input pair $(Y, \boldsymbol{X})$. For any link function $f$ in (1.1) with $\mu_0, \phi(f)$ defined in (3.2) and (3.1), and $\phi(f) > 0$[1]. We let*

$$\gamma := \left[\frac{1 - \mu_0^2}{\phi(f) + 1 - \mu_0^2} + 1\right] \Big/ 2, \quad \text{and} \quad \xi := \frac{\gamma\phi(f) + (\gamma - 1)(1 - \mu_0^2)}{(1 + \gamma)[\phi(f) + 1 - \mu_0^2]}. \tag{3.6}$$

*There exist constant $C_i$ such that when $n \geq C_1 p/\xi^2$, for Algorithm 1, we have that with probability at least $1 - 2\exp(-C_2 p)$,*

$$\left\|\boldsymbol{\beta}^t - \boldsymbol{\beta}^*\right\|_2 \leq C_3 \cdot \underbrace{\frac{\phi(f) + 1 - \mu_0^2}{\phi(f)} \cdot \sqrt{\frac{p}{n}}}_{\text{Statistical Error}} + \underbrace{\sqrt{\frac{1 - \alpha^2}{\alpha^2}} \cdot \gamma^t}_{\text{Optimization Error}}, \quad \text{for } t = 1, \ldots, T_{\max}. \tag{3.7}$$

*Here $\alpha = \langle \boldsymbol{\beta}^0, \widehat{\boldsymbol{\beta}}\rangle$, where $\widehat{\boldsymbol{\beta}}$ is the first leading eigenvector of $\mathbf{M}$.*

Note that by (3.6) we have $\gamma \in (0, 1)$. Thus, the optimization error term in (3.7) decreases at a geometric rate to zero as $t$ increases. For $T_{\max}$ sufficiently large such that the statistical error and optimization error terms in (3.7) are of the same order, we have $\left\|\boldsymbol{\beta}^{T_{\max}} - \boldsymbol{\beta}^*\right\|_2 \lesssim \sqrt{p/n}$. This statistical rate of convergence matches the rate of estimating a $p$-dimensional vector in linear regression without any quantization, and will later be shown to be optimal. This result shows that the lack of prior knowledge on the link function and the information loss from quantization do not keep our procedure from obtaining the optimal statistical rate.

### 3.4 High dimensional recovery

Next we consider the high dimensional setting where $p \gg n$ and $\boldsymbol{\beta}^*$ is sparse, i.e., $\boldsymbol{\beta}^* \in \mathbb{S}^{p-1} \cap \mathbb{B}_0(s, p)$ with $s$ being support size. Although this high dimensional estimation problem is closely

related to the well-studied sparse PCA problem, the existing works [4, 6, 17, 23, 27, 28, 31, 32] on sparse PCA do not provide a direct solution to our problem. In particular, they either lack statistical guarantees on the convergence rate of the obtained estimator [6, 23, 28] or rely on the properties of the sample covariance matrix of Gaussian data [4, 17], which are violated by the second moment estimator defined in (3.4). For the sample covariance matrix of sub-Gaussian data, [27] prove that the convex relaxation proposed by [7] achieves a suboptimal $s\sqrt{\log p/n}$ rate of convergence. Yuan and Zhang [31] propose the truncated power method, and show that it attains the optimal $\sqrt{s \log p/n}$ rate

---

**Algorithm 1** Low dimensional recovery

**Input** $\{(y_i, \mathbf{x}_i)\}_{i=1}^n$, number of iterations $T_{\max}$
  1: **Second moment estimation:** Construct $\mathbf{M}$ from samples according to (3.4).
  2: **Initialization:** Choose a random vector $\boldsymbol{\beta}^0 \in \mathbb{S}^{n-1}$
  3: **For** $t = 1, 2, \ldots, T_{\max}$ **do**
  4:    $\boldsymbol{\beta}^t \leftarrow \mathbf{M} \cdot \boldsymbol{\beta}^{t-1}$
  5:    $\boldsymbol{\beta}^t \leftarrow \boldsymbol{\beta}^t / \|\boldsymbol{\beta}^t\|_2$
  6: **end For**
**Output** $\boldsymbol{\beta}^{T_{\max}}$

---

*locally*; that is, it exhibits this rate of convergence only in a neighborhood of the true solution where $\langle \boldsymbol{\beta}^0, \boldsymbol{\beta}^* \rangle > C$ where $C > 0$ is some constant. It is well understood that for a random initialization on $\mathbb{S}^{p-1}$, such a condition fails with probability going to one as $p \to \infty$.

---

**Algorithm 2** Sparse recovery

**Input** $\{(y_i, \mathbf{x}_i)\}_{i=1}^n$, number of iterations $T_{\max}$, regularization parameter $\rho$, sparsity level $\widehat{s}$.
  1: **Second moment estimation:** Construct $\mathbf{M}$ from samples according to (3.4).
  2: **Initialization:**
  3:   $\boldsymbol{\Pi}^0 \leftarrow \underset{\boldsymbol{\Pi} \in \mathbb{R}^{p \times p}}{\operatorname{argmin}} \{ -\langle \mathbf{M}, \boldsymbol{\Pi} \rangle + \rho \|\boldsymbol{\Pi}\|_{1,1}$

  $\hspace{3cm} |\operatorname{Tr}(\boldsymbol{\Pi}) = 1, \mathbf{0} \preceq \boldsymbol{\Pi} \preceq \mathbf{I}\}$ (3.8)
  4:   $\overline{\boldsymbol{\beta}}^0 \leftarrow$ first leading eigenvector of $\boldsymbol{\Pi}^0$
  5:   $\boldsymbol{\beta}^0 \leftarrow \operatorname{trunc}(\overline{\boldsymbol{\beta}}^0, \widehat{s})$
  6:   $\boldsymbol{\beta}^0 \leftarrow \boldsymbol{\beta}^0 / \|\boldsymbol{\beta}^0\|_2$
  7: **For** $t = 1, 2, \ldots, T_{\max}$ **do**
  8:    $\boldsymbol{\beta}^t \leftarrow \operatorname{trunc}(\mathbf{M} \cdot \boldsymbol{\beta}^{t-1}, \widehat{s})$
  9:    $\boldsymbol{\beta}^t \leftarrow \boldsymbol{\beta}^t / \|\boldsymbol{\beta}^t\|_2$
 10: **end For**
**Output** $\boldsymbol{\beta}^{T_{\max}}$

---

Instead, we propose a two-stage procedure for estimating $\boldsymbol{\beta}^*$ in our setting. In the first stage, we adapt the convex relaxation proposed by [27] and use it as an initialization step, in order to obtain a good enough initial point satisfying the condition $\langle \boldsymbol{\beta}^0, \boldsymbol{\beta}^* \rangle > C$. The convex optimization problem can be easily solved by the *alternating direction method of multipliers* (ADMM) algorithm (see [3, 27] for details). Then we adapt the truncated power method. This procedure is illustrated in Algorithm 2. In particular, we define truncation operator $\operatorname{trunc}(\cdot, \cdot)$ as $[\operatorname{trunc}(\boldsymbol{\beta}, s)]_j = \mathbb{1}(j \in \mathcal{S})\beta_j$, where $\mathcal{S}$ is the index set corresponding to the top $s$ largest $|\beta_j|$. The initialization phase of our algorithm requires $O(s^2 \log p)$ samples (see below for more precise details) to succeed. As work in [2] suggests, it is unlikely that a polynomial time algorithm can avoid such dependence. However, once we are near the solution, as we show, this two-step procedure achieves the optimal error rate of $\sqrt{s \log p/n}$.

**Theorem 3.6.** *Let*

$$\kappa := \left[4(1-\mu_0^2) + \phi(f)\right] \big/ \left[4(1-\mu_0^2) + 3\phi(f)\right] < 1, \tag{3.9}$$

*and the minimum sample size be*

$$n_{\min} := C \cdot s^2 \log p \cdot \phi(f)^2 \cdot \min\{\kappa(1-\kappa^{1/2})/2, \kappa/8\} \big/ \left[(1-\mu_0^2) + \phi(f)\right]^2. \tag{3.10}$$

*Suppose $\rho = C\left[\phi(f) + (1-\mu_0^2)\right]\sqrt{\log p/n}$ with a sufficiently large constant $C$, where $\phi(f)$ and $\mu_0$ are specified in (3.2) and (3.5). Meanwhile, assume the sparsity parameter $\widehat{s}$ in Algorithm 2 is set to be $\widehat{s} = C'' \max\left\{\lceil 1/(\kappa^{-1/2}-1)^2\rceil, 1\right\} \cdot s^*$. For $n \geq n_{\min}$ with $n_{\min}$ defined in (3.10), we have*

$$\|\boldsymbol{\beta}^t - \boldsymbol{\beta}^*\|_2 \leq C \cdot \underbrace{\frac{\left[\phi(f) + (1-\mu_0^2)\right]^{\frac{5}{2}}(1-\mu_0^2)^{\frac{1}{2}}}{\phi(f)^3} \cdot \sqrt{\frac{s \log p}{n}}}_{\text{Statistical Error}} + \underbrace{\kappa^t \cdot \sqrt{\min\{(1-\kappa^{1/2})/2, 1/8\}}}_{\text{Optimization Error}}$$

(3.11)

*with high probability. Here $\kappa$ is defined in* (3.9)*.*

The first term on the right-hand side of (3.11) is the statistical error while the second term gives the optimization error. Note that the optimization error decays at a geometric rate since $\kappa < 1$. For $T_{\max}$

sufficiently large, we have

$$\left\|\boldsymbol{\beta}^{T_{\max}} - \boldsymbol{\beta}^*\right\|_2 \lesssim \sqrt{s \log p/n}.$$

In the sequel, we show that the right-hand side gives the optimal statistical rate of convergence for a broad model class under the high dimensional setting with $p \gg n$.

## 3.5   Minimax lower bound

We establish the minimax lower bound for estimating $\boldsymbol{\beta}^*$ in the model defined in (1.1). In the sequel we define the family of link functions that are Lipschitz continuous and are bounded away from $\pm 1$. Formally, for any $m \in (0,1)$ and $L > 0$, we define

$$\mathcal{F}(m, L) := \left\{ f : |f(z)| \leq 1 - m, \quad |f(z) - f(z')| \leq L|z - z'|, \quad \text{for all } z, z' \in \mathbb{R} \right\}. \quad (3.12)$$

Let $\mathcal{X}_f^n := \{(y_i, \mathbf{x}_i)\}_{i=1}^n$ be the $n$ i.i.d. realizations of $(Y, \boldsymbol{X})$, where $\boldsymbol{X}$ follows $\mathcal{N}(\mathbf{0}, \mathbf{I}_p)$ and $Y$ satisfies (1.1) with link function $f$. Correspondingly, we denote the estimator of $\boldsymbol{\beta}^* \in \mathcal{B}$ to be $\widehat{\boldsymbol{\beta}}(\mathcal{X}_f^n)$, where $\mathcal{B}$ is the domain of $\boldsymbol{\beta}^*$. We define the minimax risk for estimating $\boldsymbol{\beta}^*$ as

$$\mathcal{R}(n, m, L, \mathcal{B}) := \inf_{f \in \mathcal{F}(m,L)} \inf_{\widehat{\boldsymbol{\beta}}(\mathcal{X}_f^n)} \sup_{\boldsymbol{\beta}^* \in \mathcal{B}} \mathbb{E} \left\|\widehat{\boldsymbol{\beta}}(\mathcal{X}_f^n) - \boldsymbol{\beta}^*\right\|_2. \quad (3.13)$$

In the above definition, we not only take the infimum over all possible estimators $\widehat{\boldsymbol{\beta}}$, but also all possible link functions in $\mathcal{F}(m, L)$. For a fixed $f$, our formulation recovers the standard definition of minimax risk [30]. By taking the infimum over all link functions, our formulation characterizes the minimax lower bound under the least challenging $f$ in $\mathcal{F}(m, L)$. In the sequel we prove that our procedure attains such a minimax lower bound for the least challenging $f$ given any unknown link function in $\mathcal{F}(m, L)$. That is to say, even when $f$ is unknown, our estimation procedure is as accurate as in the setting where we are provided the least challenging $f$, and the achieved accuracy is not improvable due to the information-theoretic limit. The following theorem establishes the minimax lower bound in the high dimensional setting.

**Theorem 3.7.** *Let* $\mathcal{B} = \mathbb{S}^{p-1} \cap \mathbb{B}_0(s, p)$. *We assume that* $n > m(1-m)/(2L^2)^2 \left[ Cs \log(p/s)/2 - \log 2 \right]$. *For any* $s \in (0, p/4]$, *the minimax risk defined in* (3.13) *satisfies*

$$\mathcal{R}(n, m, L, \mathcal{B}) \geq C' \cdot \frac{\sqrt{m(1-m)}}{L} \cdot \sqrt{\frac{s \log(p/s)}{n}}.$$

*Here* $C$ *and* $C'$ *are absolute constants, while* $m$ *and* $L$ *are defined in* (3.12).

Theorem 3.7 establishes the minimax optimality of the statistical rate attained by our procedure for $p \gg n$ and $s$-sparse $\boldsymbol{\beta}^*$. In particular, for arbitrary $f \in \mathcal{F}(m, L) \cap \{f : \phi(f) > 0\}$, the estimator $\widehat{\boldsymbol{\beta}}$ attained by Algorithm 2 is minimax-optimal in the sense that its $\sqrt{s \log p/n}$ rate of convergence is not improvable, even when the information on the link function $f$ is available. For general $\boldsymbol{\beta}^* \in \mathbb{R}^p$, one can show the best possible convergence rate is $\Omega(\sqrt{m(1-m)p/n}/L)$ by setting $s = p/4$ in Theorem 3.7.

It is worth to note that our lower bound becomes trivial for $m = 0$, i.e., there exists some $z$ such that $|f(z)| = 1$. One example is the noiseless one-bit compressed sensing for which we have $f(z) = \text{sign}(z)$. In fact, for noiseless one-bit compressed sensing, the $\sqrt{s \log p/n}$ rate is not optimal. For example, the Jacques et al. [14] provide an algorithm (with exponential running time) that achieves rate $s \log p/n$. Understanding such a rate transition phenomenon for link functions with zero margin, i.e., $m = 0$ in (3.12), is an interesting future direction.

## 3.6   Numerical results

We now turn to the numerical results that support our theory. For the three models introduced in §2, we apply Algorithm 1 and Algorithm 2 to do parameter estimation in the classic and high dimensional regimes. Our simulations are based on synthetic data. For classic recovery, $\boldsymbol{\beta}^*$ is randomly chosen from $\mathbb{S}^{p-1}$; for sparse recovery, we set $\beta_j^* = s^{-1/2} \mathbb{1}(j \in \mathcal{S})$ for all $j \in [p]$, where $\mathcal{S}$ is a random index subset of $[p]$ with size $s$. In Figure 1, as predicted by Theorem 3.5, we observe that the same

$\sqrt{p/n}$ leads to nearly identical estimation error. Figure 2 demonstrates similar results for the predicted rate $\sqrt{s \log p/n}$ of sparse recovery and thus validates Theorem 3.6.

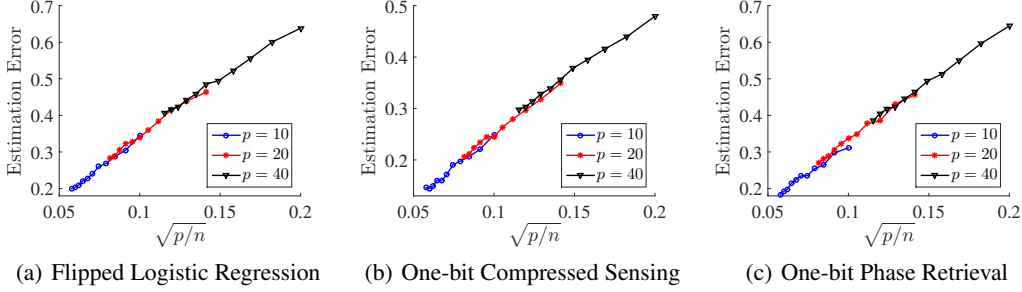

| (a) Flipped Logistic Regression | (b) One-bit Compressed Sensing | (c) One-bit Phase Retrieval |

Figure 1: Estimation error of low dimensional recovery. (a) $p_e = 0.1$. (b) $\delta^2 = 0.1$. (c) $\theta = 1$.

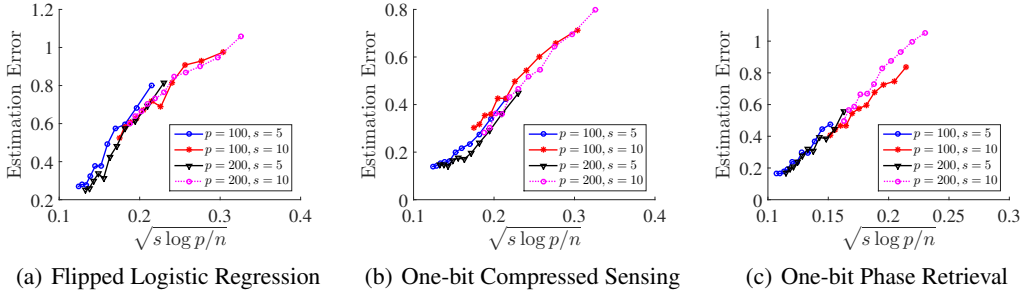

| (a) Flipped Logistic Regression | (b) One-bit Compressed Sensing | (c) One-bit Phase Retrieval |

Figure 2: Estimation error of sparse recovery. (a) $p_e = 0.1$. (b) $\delta^2 = 0.1$. (c) $\theta = 1$.

## 4  Discussion

*Sample complexity.* In high dimensional regime, while our algorithm achieves optimal convergence rate, the sample complexity we need is $\Omega(s^2 \log p)$. The natural question is whether it can be reduced to $O(s \log p)$. We note that breaking the barrier $s^2 \log p$ is challenging. Consider a simpler problem *sparse phase retrieval* where $y_i = |\langle \mathbf{x}_i, \boldsymbol{\beta}^* \rangle|$, with a fairly extensive body of literature, the state-of-the-art efficient algorithms (i.e., with polynomial running time) for recovering sparse $\boldsymbol{\beta}^*$ requires sample complexity $\Omega(s^2 \log p)$ [10]. It remains open to show whether it's possible to do consistent sparse recovery with $O(s \log p)$ samples by any polynomial time algorithms.

## Acknowledgment

XY and CC would like to acknowledge NSF grants 1056028, 1302435 and 1116955. This research was also partially supported by the U.S. Department of Transportation through the Data-Supported Transportation Operations and Planning (D-STOP) Tier 1 University Transportation Center. HL is grateful for the support of NSF CAREER Award DMS1454377, NSF IIS1408910, NSF IIS1332109, NIH R01MH102339, NIH R01GM083084, and NIH R01HG06841. ZW was partially supported by MSR PhD fellowship while this work was done.

## Footnotes

[1]Recall that we have an analogous treatment and thus results for $\phi(f) < 0$.

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
