[Supplementary Material]

# A  Proofs

In this section, we provide the proofs for our main results. First we characterize the implications of our general framework for the models in §2. We then establish the statistical convergence rates of the proposed procedure and the corresponding minimax lower bounds.

## A.1  Proof of Lemma 3.2

Let $\boldsymbol{X}$ and $\boldsymbol{X}'$ be two independent random vectors following $\mathcal{N}(\mathbf{0}, \mathbf{I}_p)$. Let $Y$ and $Y'$ be two binary responses that depend on $\boldsymbol{X}, \boldsymbol{X}'$ via (1.1). Then we have

$$\mathbb{E}(\mathbf{M}) = \mathbb{E}\big[(Y - Y')^2 (\boldsymbol{X} - \boldsymbol{X}')(\boldsymbol{X} - \boldsymbol{X}')^\top\big].$$

Note that $(Y - Y')^2$ is a binary random variable taking values in $\{0, 4\}$. We have

$$\begin{aligned}
\mathbb{E}\big[(Y - Y')^2 | \boldsymbol{X} = \mathbf{x}, \boldsymbol{X}' = \mathbf{x}'\big] &= 4 \cdot \mathbb{P}\big[(Y - Y')^2 = 4 | \boldsymbol{X} = \mathbf{x}, \boldsymbol{X}' = \mathbf{x}'\big] \\
&= 4 \cdot \mathbb{P}(Y = 1 | \boldsymbol{X} = \mathbf{x}) \cdot \mathbb{P}(Y' = -1 | \boldsymbol{X}' = \mathbf{x}') + 4 \cdot \mathbb{P}(Y' = 1 | \boldsymbol{X}' = \mathbf{x}') \cdot \mathbb{P}(Y = -1 | \boldsymbol{X} = \mathbf{x}) \\
&= 2 - 2 f(\langle \boldsymbol{x}, \boldsymbol{\beta}^* \rangle) f(\langle \boldsymbol{x}', \boldsymbol{\beta}^* \rangle). \quad\quad\quad (A.1)
\end{aligned}$$

There exists some rotation matrix $\mathbf{Q} \in \mathbb{R}^{p \times p}$ such that $\mathbf{Q}\boldsymbol{\beta}^* = \boldsymbol{e}_1 := [1, 0, \ldots, 0]^\top$. Let $\overline{\boldsymbol{X}} := \mathbf{Q}\boldsymbol{X}$ and $\overline{\boldsymbol{X}}' := \mathbf{Q}\boldsymbol{X}'$. Then we have

$$\mathbb{E}\big[(Y - Y')^2 | \boldsymbol{X} = \mathbf{x}, \boldsymbol{X}' = \mathbf{x}'\big] = \mathbb{E}\big[(Y - Y')^2 | \overline{\boldsymbol{X}} = \mathbf{Q}\mathbf{x}, \overline{\boldsymbol{X}}' = \mathbf{Q}\mathbf{x}'\big] = 2 - 2 \cdot f(\bar{x}_1) \cdot f(\bar{x}'_1),$$

where $\bar{x}_1$ and $\bar{x}'_1$ denote the first entries of $\bar{\mathbf{x}} := \mathbf{Q}\mathbf{x}$ and $\bar{\mathbf{x}}' := \mathbf{Q}\mathbf{x}'$ respectively. Note that $\overline{\boldsymbol{X}}$ and $\overline{\boldsymbol{X}}'$ also follow $\mathcal{N}(\mathbf{0}, \mathbf{I}_p)$ since symmetric Gaussian distribution is rotation invariant. Then we have

$$\begin{aligned}
\mathbb{E}(\mathbf{M}) &= \mathbb{E}\left\{ \big[2 - 2 f(\overline{\boldsymbol{X}}_1) f(\overline{\boldsymbol{X}}'_1)\big] (\boldsymbol{X} - \boldsymbol{X}')(\boldsymbol{X} - \boldsymbol{X}')^\top \right\} \\
&= \mathbf{Q}^\top \mathbb{E}\left\{ \big[2 - 2 f(\overline{\boldsymbol{X}}_1) f(\overline{\boldsymbol{X}}'_1)\big] (\overline{\boldsymbol{X}} - \overline{\boldsymbol{X}}')(\overline{\boldsymbol{X}} - \overline{\boldsymbol{X}}')^\top \right\} \mathbf{Q} \\
&= 4\mathbf{Q}^\top \big[ (\mu_1^2 - \mu_0 \mu_2 + \mu_0^2) \cdot \boldsymbol{e}_1 \boldsymbol{e}_1^\top + (1 - \mu_0^2) \cdot \mathbf{I}_p \big] \mathbf{Q} = 4\phi(f) \cdot \boldsymbol{\beta}^* \boldsymbol{\beta}^{*\top} + 4(1 - \mu_0^2) \cdot \mathbf{I}_p.
\end{aligned}$$

The third equality is from the definitions of $\mu_0, \mu_1, \mu_2$ in (3.2) and the last equality is from (3.1).

## A.2  Proof of Lemma 3.4

**Flipped logistic regression.** For flipped logistic regression, the link function $f$ is defined in (2.1), where $\zeta$ is the intercept. For $\zeta = 0$, we have

$$f(z) = \frac{e^z - 1}{e^z + 1} + 2p_\mathrm{e} \cdot \frac{1 - e^z}{1 + e^z}.$$

Note that $f$ is odd. Hence, by (3.2) we have $\mu_0 = \mu_2 = 0$. Meanwhile, from Stein's lemma, we have

$$\mu_1 = \mathbb{E}[f'(z)] = \mathbb{E}\left[ (1 - 2p_\mathrm{e}) \cdot \frac{2e^z}{(1 + e^z)^2} \right] = (1 - 2p_\mathrm{e}) \cdot \mathbb{E}\frac{2e^z}{(1 + e^z)^2}.$$

We thus have $\phi(f) = \mu_1^2 \geq C(1 - 2p_\mathrm{e})^2$ for some constant $C$.

**Robust one-bit compressed sensing.** Recall in robust one-bit compressed sensing, we have

$$f(z) = 2 \cdot \mathbb{P}(z + \epsilon > 0) - 1,$$

where $\epsilon \sim \mathcal{N}(0, \sigma^2)$ is the noise term in (2.2). In particular, note that

$$f(z) + f(-z) = 2 \cdot \big[\mathbb{P}(\epsilon > z) + \mathbb{P}(\epsilon > -z)\big] - 2 = 0.$$

Hence, $f(z)$ is an odd function, which implies $\mu_0 = \mu_2 = 0$ by (3.2). For $\mu_1$ defined in (3.2), we have

$$\mu_1 = \mathbb{E}[f(z)z] = \mathbb{E}\big\{ \big[2 \cdot \mathbb{P}(\epsilon > -z) - 1\big] z \big\} = \mathbb{E}\big[\mathbb{P}(|\epsilon| < |z|)|z|\big] \geq \mathbb{E}\left\{ \big[1 - 2e^{-z^2/(2\sigma^2)}\big] |z| \right\} \quad (A.2)$$

$$= \mathbb{E}(|z|) - \int_{-\infty}^{\infty} \frac{2}{\sqrt{2\pi}} e^{-\frac{u^2}{2\sigma^2}} e^{-\frac{u^2}{2}} |u| \mathrm{d}u = \mathbb{E}(|z|) \left(1 - 2\frac{\sigma^2}{1 + \sigma^2}\right) = \mathbb{E}(|z|) \frac{1 - \sigma^2}{1 + \sigma^2}.$$

Here the inequality is from the fact that $\mathbb{P}(|\epsilon| < |z|) \geq 1 - 2e^{-\frac{z^2}{2\sigma^2}}$ since $\epsilon \sim \mathcal{N}(0, \sigma^2)$. For $\sigma^2 < 1/2$, we have

$$\phi(f) = \mu_1^2 \geq C \left(\frac{1 - \sigma^2}{1 + \sigma^2}\right)^2,$$

where $C = \mathbb{E}(|z|)$ with $z \sim \mathcal{N}(0, 1)$. For $\sigma^2 \geq 1/2$, rather than applying $\mathbb{P}(|\epsilon| < |z|) \geq 1 - 2e^{-\frac{z^2}{2\sigma^2}}$ in the inequality of (A.2), we apply $\mathbb{P}(|\epsilon| < |z|) \geq \frac{2}{\sqrt{2\pi}\sigma}e^{-\frac{z^2}{2\sigma^2}}|z|$ since $\epsilon \sim \mathcal{N}(0, \sigma^2)$. We then obtain

$$\mu_1 \geq \mathbb{E}\left[\frac{2}{\sqrt{2\pi}\sigma}e^{-\frac{z^2}{2\sigma^2}}z^2\right] = \frac{2}{\sqrt{2\pi}\sigma}\int_{-\infty}^{\infty}\frac{1}{\sqrt{2\pi}}e^{-\frac{u^2}{2\sigma^2}}e^{-\frac{u^2}{2}}u^2\mathrm{d}u \geq \frac{C'}{\sigma}\left(\frac{\sigma^2}{1 + \sigma^2}\right)^{\frac{3}{2}}.$$

Finally, for $\sigma^2 \geq 1/2$ we have

$$\phi(f) \geq \frac{C'\sigma^4}{(1 + \sigma^2)^3}.$$

**One-bit phase retrieval.** For the one-bit phase retrieval model, the major difference from the previous two models is that $f(z)$ is even, which results in $\mu_1 = 0$. By the definition in (3.2), we have

$$\mu_0 = \mathbb{E}[f(z)] = \mathbb{P}(|z| \geq \theta) - \mathbb{P}(|z| < \theta),$$

and

$$\mu_2 = \mathbb{E}\left[f(z)z^2\right] = \mathbb{P}(|z| \geq \theta)\mathbb{E}\left(z^2 \mid |z| \geq \theta\right) - \mathbb{P}(|z| < \theta)\mathbb{E}\left(z^2 \mid |z| < \theta\right).$$

For notational simplicity, we define $p_1 = \mathbb{P}(|z| \geq \theta)$. We have

$$\phi(f) = \mu_0(\mu_0 - \mu_2) = 2p_1(2p_1 - 1)\left[1 - \mathbb{E}\left(z^2 \mid |z| > \theta\right)\right], \tag{A.3}$$

where the second equality follows from the fact that

$$\mathbb{P}(|z| \geq \theta) + \mathbb{P}(|z| < \theta) = 1, \tag{A.4}$$

and

$$\mathbb{P}(|z| \geq \theta)\mathbb{E}\left(z^2 \mid |z| \geq \theta\right) + \mathbb{P}(|z| < \theta)\mathbb{E}\left(z^2 \mid |z| < \theta\right) = \mathbb{E}(z^2) = 1. \tag{A.5}$$

By (A.4) and (A.5) we have $p_1 > 0$ and $\mathbb{E}\left(z^2 \mid |z| \geq \theta\right) > 1$ for $\theta > 0$. Hence, for $\theta < \theta_\mathrm{m}$ with $\theta_\mathrm{m}$ being the median of $|z|$ with $z \sim \mathcal{N}(0, 1)$, we have $p_1 \geq 1/2$, which further implies $\phi(f) < 0$ by (A.3). Otherwise we have $\phi(f) > 0$. Thus, we have $\mathrm{sign}[\phi(f)] = \mathrm{sign}(\theta - \theta_\mathrm{m})$.

In the following we establish a lower bound for $|\phi(f)|$. Note that

$$\mathbb{E}\left(z^2 \mid |z| \geq \theta\right) = \frac{2}{p_1}\int_{\theta}^{+\infty}\frac{1}{\sqrt{2\pi}}e^{-\frac{z^2}{2}}z^2dz = \frac{2\theta}{p_1\sqrt{2\pi}}e^{-\frac{\theta^2}{2}} + 1. \tag{A.6}$$

Plugging (A.6) into (A.3) yields

$$\phi(f) = -2(2p_1 - 1)\frac{2\theta}{\sqrt{2\pi}}e^{-\frac{\theta^2}{2}}. \tag{A.7}$$

For $0 < \theta < \theta_\mathrm{m}$, which implies $p_1 \geq 1/2$, we have

$$p_1 - \frac{1}{2} = 2\int_{\theta}^{\theta_\mathrm{m}}\frac{1}{\sqrt{2\pi}}e^{-\frac{z^2}{2}}dz \geq \frac{2}{\sqrt{2\pi}}e^{-\frac{\theta_\mathrm{m}^2}{2}}(\theta_\mathrm{m} - \theta). \tag{A.8}$$

By plugging (A.6) into (A.7), we have

$$|\phi(f)| \geq \frac{8}{\sqrt{2\pi}}e^{-\frac{\theta_\mathrm{m}^2}{2}}(\theta_\mathrm{m} - \theta)\frac{2\theta}{\sqrt{2\pi}}e^{-\frac{\theta^2}{2}} \geq C\theta(\theta_\mathrm{m} - \theta)e^{-\frac{\theta^2}{2}}. \tag{A.9}$$

For $\theta > \theta_\mathrm{m}$, which implies $p_1 < 1/2$, similarly to (A.8), we have

$$\frac{1}{2} - p_1 = 2\int_{\theta_\mathrm{m}}^{\theta}\frac{1}{\sqrt{2\pi}}e^{-\frac{z^2}{2}}dx \geq \frac{2}{\sqrt{2\pi}}e^{-\frac{\theta^2}{2}}(\theta - \theta_\mathrm{m}). \tag{A.10}$$

Thus, we conclude that

$$|\phi(f)| \geq C'\theta(\theta - \theta_\mathrm{m})e^{-\theta^2}.$$

## A.3 Proof of Theorem 3.5

Let $\widehat{\boldsymbol{\beta}}$ be the top eigenvector of $\mathbf{M}$ and $\widehat{\lambda}_1, \widehat{\lambda}_2$ be the first and second largest eigenvalues of $\mathbf{M}$. We use $\lambda_1, \lambda_2$ to denote the first and second largest eigenvalues of $\mathbb{E}(\mathbf{M})$. From Lemma 3.2, we already know that

$$\lambda_1 = 4\phi(f) + 4(1 - \mu_0^2), \quad \text{and} \ \lambda_2 = 4(1 - \mu_0^2).$$

By the triangle inequality, we have

$$\left\|\boldsymbol{\beta}^t - \boldsymbol{\beta}^*\right\|_2 \le \left\|\boldsymbol{\beta}^* - \widehat{\boldsymbol{\beta}}\right\|_2 + \left\|\boldsymbol{\beta}^t - \widehat{\boldsymbol{\beta}}\right\|_2.$$

The first term on the right hand side is the statistical error and the second term is the optimization error. From standard analysis of the power method, we have

$$\left\|\boldsymbol{\beta}^t - \widehat{\boldsymbol{\beta}}\right\|_2 \le \sqrt{\frac{1 - \alpha^2}{\alpha^2}} \cdot \left(\widehat{\lambda}_2 / \widehat{\lambda}_1\right)^t,$$

where $\alpha = \langle \boldsymbol{\beta}^0, \widehat{\boldsymbol{\beta}} \rangle$. By the definition in (3.4), $\mathbf{M}$ is the sample covariance matrix of $n/2$ independent realizations of the random vector $(Y - Y')(\boldsymbol{X} - \boldsymbol{X}') \in \mathbb{R}^p$. Since $\boldsymbol{X}$ is Gaussian and $Y$ is bounded, $(Y - Y')(\boldsymbol{X} - \boldsymbol{X}')$ is sub-Gaussian. By standard concentration results (see e.g. Theorem 5.39 in [26]), there some constants $C, C_1$ such that for any $t \ge 0$, with probability at least $1 - 2e^{-Ct^2}$,

$$\|\mathbf{M} - \mathbb{E}(\mathbf{M})\|_2 \le \max(\delta, \delta^2)\|\mathbb{E}(\mathbf{M})\|_2,$$

where $\delta = C_1 \sqrt{\frac{p}{n}} + \frac{t}{\sqrt{n}}$. We let $t = \sqrt{p}$, then for any $\xi \in (0, 1)$, we have that $\|\mathbf{M} - \mathbb{E}(\mathbf{M})\|_2 \le \xi \|\mathbb{E}(\mathbf{M})\|_2$ when $n \ge C_2 p/\xi^2$ for sufficiently large constant $C_2$. Conditioning on $\|\mathbf{M} - \mathbb{E}(\mathbf{M})\|_2 \le \xi \|\mathbb{E}(\mathbf{M})\|_2$, from Weyl's inequality, we have

$$\widehat{\lambda}_1 \ge 4(1 - \xi)\big[\phi(f) + 1 - \mu_0^2\big], \quad \text{and} \ \widehat{\lambda}_2 \le 4\xi\phi(f) + 4(1 + \xi)(1 - \mu_0^2).$$

Furthermore, for any $\gamma \in \big((1 - \mu_0^2)/\big[\phi(f) + 1 - \mu_0^2\big], 1\big)$, by restricting

$$\xi \le \frac{\gamma\phi(f) + (\gamma - 1)(1 - \mu_0^2)}{(1 + \gamma)\big[\phi(f) + 1 - \mu_0^2\big]}, \tag{A.11}$$

we have

$$\left\|\boldsymbol{\beta}^t - \widehat{\boldsymbol{\beta}}\right\|_2 \le \sqrt{\frac{1 - \alpha^2}{\alpha^2}} \cdot \gamma^t.$$

Now we turn to the statistical error. By Wedin's sin theorem, for some positive constant $C > 0$, we have

$$\sin \angle\big(\boldsymbol{\beta}^*, \widehat{\boldsymbol{\beta}}\big) \le C \cdot \frac{\xi\|\mathbb{E}(\mathbf{M})\|_2}{\lambda_1 - \lambda_2}. \tag{A.12}$$

Elementary calculation yields

$$\left\|\widehat{\boldsymbol{\beta}} - \boldsymbol{\beta}^*\right\|_2 = 2\sin\big[\angle\big(\boldsymbol{\beta}^*, \widehat{\boldsymbol{\beta}}\big)/2\big] \le \sqrt{2}\sin \angle\big(\boldsymbol{\beta}^*, \widehat{\boldsymbol{\beta}}\big). \tag{A.13}$$

As $\xi \lesssim \sqrt{p/n}$, combining (A.12) and (A.13), we have

$$\left\|\widehat{\boldsymbol{\beta}} - \boldsymbol{\beta}^*\right\|_2 \lesssim \frac{\phi(f) + 1 - \mu_0^2}{\phi(f)} \cdot \sqrt{\frac{p}{n}}.$$

Putting all pieces together, we conclude that if $\xi$ satisfies (A.11) and $n \gtrsim p/\xi^2$, then we have that with probability at least $1 - 2e^{-Cp}$,

$$\left\|\boldsymbol{\beta}^t - \boldsymbol{\beta}^*\right\|_2 \le C \cdot \frac{\phi(f) + 1 - \mu_0^2}{\phi(f)} \cdot \sqrt{\frac{p}{n}} + \sqrt{\frac{1 - \alpha^2}{\alpha^2}} \cdot \gamma^t.$$

as required.

## A.4 Proof of Theorem 3.6

The analysis of Algorithm 2 follows from a combination of [27] (for the initialization via convex relaxation) and [31] (for the original truncated power method). Recall that $\kappa$ is defined in (3.9).

Assume the initialization $\boldsymbol{\beta}^0$ is $\widehat{s}$-sparse with $\|\boldsymbol{\beta}^0\|_2 = 1$, and satisfies

$$\left\|\boldsymbol{\beta}^0 - \boldsymbol{\beta}^*\right\|_2 \leq C \min\left\{\sqrt{\kappa(1-\kappa^{1/2})/2}, \sqrt{2\kappa}/4\right\}, \tag{A.14}$$

for $\widehat{s} = C' \max\left\{\left\lceil 1/(\kappa^{-1/2}-1)^2\right\rceil, 1\right\} \cdot s$. Theorem 1 of [31] implies that

$$\left\|\boldsymbol{\beta}^t - \boldsymbol{\beta}^*\right\|_2 \leq C'' \cdot \frac{\left[\phi(f) + (1-\mu_0^2)\right]^{\frac{5}{2}}(1-\mu_0^2)^{\frac{1}{2}}}{\phi^3} \cdot \sqrt{\frac{s \log p}{n}} + \kappa^t \cdot \sqrt{\min\left\{(1-\kappa^{1/2})/2, 1/8\right\}}$$

with high probability. Therefore, we only need to prove the initialization $\boldsymbol{\beta}^0$ obtained in Algorithm 2 satisfies the condition in (A.14).

Corollary 3.3 of [27] shows that the minimizer to the minimization problem in line 3 of Algorithm 2 satisfies

$$\left\|\boldsymbol{\Pi}^0 - \boldsymbol{\beta}^* \cdot (\boldsymbol{\beta}^*)^\top\right\|_2 \leq C \cdot \frac{\phi(f) + (1-\mu_0^2)}{\phi} \cdot s\sqrt{\frac{\log p}{n}}$$

with high probability. Corollary 3.2 of [27] implies, the first eigenvector of $\boldsymbol{\Pi}^0$, denoted as $\overline{\boldsymbol{\beta}}^0$, satisfies

$$\left\|\overline{\boldsymbol{\beta}}^0 - \boldsymbol{\beta}^*\right\|_2 \leq C' \cdot \frac{\phi(f) + (1-\mu_0^2)}{\phi} \cdot s\sqrt{\frac{\log p}{n}}$$

with the same probability. However, $\overline{\boldsymbol{\beta}}^0$ is not necessarily $\widehat{s}$-sparse. Using Lemma 12 of [31], we obtain that the truncate step in lines 12-15 of Algorithm 2 ensures that $\boldsymbol{\beta}^0$ is $\widehat{s}$-sparse and also satisfies

$$\left\|\boldsymbol{\beta}^0 - \boldsymbol{\beta}^*\right\|_2 \leq \left(1 + 2\sqrt{\widehat{s}/s}\right) \cdot \left\|\overline{\boldsymbol{\beta}}^0 - \boldsymbol{\beta}^*\right\|_2 \leq 3\|\overline{\boldsymbol{\beta}}^0 - \boldsymbol{\beta}^*\|_2,$$

where the last inequality follows from our assumption that $\widehat{s} \geq s$. Therefore, we only have to set $n$ to be sufficiently large such that

$$\left\|\boldsymbol{\beta}^0 - \boldsymbol{\beta}^*\right\|_2 \leq C' \cdot \frac{\phi(f) + (1-\mu_0^2)}{\phi(f)} \cdot s\sqrt{\frac{\log p}{n}} \leq C\min\left\{\sqrt{\kappa(1-\kappa^{1/2})/2}, \sqrt{2\kappa}/4\right\},$$

which is ensured by setting $n \geq n_{\min}$ with

$$n_{\min} = C' \cdot s^2 \log p \cdot \phi(f)^2 \cdot \min\left\{\kappa(1-\kappa^{1/2})/2, \kappa/8\right\} / \left[(1-\mu_0^2) + \phi(f)\right]^2,$$

as specified in our assumption. Thus we conclude the proof.

## A.5    Proof of Theorem 3.7

The proof of the minimax lower bound follows from the basic idea of reducing an estimation problem to a testing problem, and then invoking Fano's inequality to lower bound the testing error. We first introduce a finite packing set for $\mathbb{S}^{p-1} \cap \mathbb{B}_0(s, p)$.

**Lemma A.1.** *Consider the set $\{0,1\}^p$ equipped with Hamming distance $\delta$. For $s \leq p/4$, there exists a finite subset $\mathcal{Q} \subset \{0,1\}^p$ such that*

$$\delta(\boldsymbol{\theta}, \boldsymbol{\theta}') > s/2, \ \ \forall (\boldsymbol{\theta}, \boldsymbol{\theta}') \in \mathcal{Q} \times \mathcal{Q} \ \text{and} \ \boldsymbol{\theta} \neq \boldsymbol{\theta}', \ \|\boldsymbol{\theta}\|_0 = s, \ \text{for all } \boldsymbol{\theta} \in \mathcal{Q}.$$

*The cardinality of such a set satisfies*

$$\log(|\mathcal{Q}|) \geq 8/3 \cdot s \log(p/s).$$

*Proof.* See the proof of Lemma 4.10 in [18]. $\qquad\square$

We use $\mathcal{Q}(p, s)$ to denote the finite set specified in Lemma A.1. For $\xi < 1$, we construct a finite subset $\overline{\mathcal{Q}}(p, s, \xi) \subset \mathbb{S}^{p-1} \cap \mathbb{B}_0(s, p)$ as

$$\overline{\mathcal{Q}}(p, s, \xi) := \left\{\boldsymbol{\beta} \in \mathbb{R}^p : \boldsymbol{\beta} = \left(\sqrt{1-\xi^2}, \frac{\xi}{\sqrt{s-1}} \cdot \boldsymbol{w}\right), \ \text{where } \boldsymbol{w} \in \mathcal{Q}(p-1, s-1)\right\}. \tag{A.15}$$

It is easy to verify that set $\overline{\mathcal{Q}}(p, s, \xi)$ has the following properties:

- For any $\boldsymbol{\theta} \in \overline{\mathcal{Q}}(p, s, \xi)$, it holds that $\|\boldsymbol{\theta}\|_2 = 1$ and $\|\boldsymbol{\theta}\|_0 = s$.

- For distinct $\boldsymbol{\theta}, \boldsymbol{\theta}' \in \overline{\mathcal{Q}}(p, s, \xi)$, $\|\boldsymbol{\theta} - \boldsymbol{\theta}'\|_2 \geq \sqrt{2}\xi/2$ and $\|\boldsymbol{\theta} - \boldsymbol{\theta}'\|_2 \leq \sqrt{2}\xi$.
- $\log|\overline{\mathcal{Q}}(p, s, \xi)| \geq Cs\log(p/s)$ for some positive constant $C$.

In order to derive lower bound of $\mathcal{R}(n, m, L, \mathcal{B})$ with $\mathcal{B} = \mathbb{S}^{p-1} \cap \boldsymbol{B}_0(s, p)$, we assume that the infimum over $f$ in (3.13) is obtained for a certain $f^* \in \mathcal{F}(m, L)$, namely

$$\mathcal{R}(n, m, L, \mathcal{B}) = \inf_{\widehat{\boldsymbol{\beta}} \in \mathbb{S}^{p-1}} \sup_{\boldsymbol{\beta} \in \mathbb{S}^{p-1} \cap \boldsymbol{B}_0(s,p)} \mathbb{E}\big\|\widehat{\boldsymbol{\beta}}(\mathcal{X}_{f^*}^n) - \boldsymbol{\beta}\big\|_2 \geq \inf_{\widehat{\boldsymbol{\beta}} \in \mathbb{S}^{p-1}} \sup_{\boldsymbol{\beta} \in \overline{\mathcal{Q}}(p,s,\xi)} \mathbb{E}\big\|\widehat{\boldsymbol{\beta}}(\mathcal{X}_{f^*}^n) - \boldsymbol{\beta}\big\|_2.$$

Note that for any $\xi > 0$, we have $\|\boldsymbol{\beta}_1 - \boldsymbol{\beta}_2\|_2 \geq \frac{\sqrt{2}}{2}\xi$ for any two distinct vectors $(\boldsymbol{\beta}_1, \boldsymbol{\beta}_2)$ in $\overline{\mathcal{Q}}(p, s, \xi)$. Therefore, we are in a position to apply standard minimax risk lower bound. Following Lemma 3 in Yu [30], we obtain

$$\inf_{\widehat{\boldsymbol{\beta}} \in \mathbb{S}^{p-1}} \sup_{\boldsymbol{\beta} \in \overline{\mathcal{Q}}(p,s,\xi)} \mathbb{E}\big\|\widehat{\boldsymbol{\beta}}(\mathcal{X}_{f^*}^n) - \boldsymbol{\beta}\big\|_2 \geq \frac{\sqrt{2}}{4}\xi\left(1 - \frac{\max_{\boldsymbol{\beta}, \boldsymbol{\beta}' \in \overline{\mathcal{Q}}(p,s,\xi)} D_{KL}(P_{\boldsymbol{\beta}'}\|P_{\boldsymbol{\beta}}) + \log 2}{\log|\overline{\mathcal{Q}}(p, s, \xi)|}\right).$$
(A.16)

In the following, we derive an upper bound for the term involving KL divergence on the right hand side of the above inequality. For any $\boldsymbol{\beta}, \boldsymbol{\beta}' \in \overline{\mathcal{Q}}(p, s, \xi)$, we have

$$D_{KL}(P_{\boldsymbol{\beta}'}\|P_{\boldsymbol{\beta}}) \leq n \cdot D_{KL}\big[P_{\boldsymbol{\beta}'}(Y, \boldsymbol{X})\|P_{\boldsymbol{\beta}}(Y, \boldsymbol{X})\big] = n \cdot \mathbb{E}_{\boldsymbol{X}}\big\{D_{KL}\big[P_{\boldsymbol{\beta}'}(Y|\boldsymbol{X})\|P_{\boldsymbol{\beta}}(Y|\boldsymbol{X})\big]\big\}$$

$$= \frac{1}{2}n \cdot \mathbb{E}_{\boldsymbol{X}}\left\{[1 + f^*(\boldsymbol{X}^\top\boldsymbol{\beta})]\log\frac{1 + f^*(\boldsymbol{X}^\top\boldsymbol{\beta})}{1 + f^*(\boldsymbol{X}^\top\boldsymbol{\beta}')} + [1 - f^*(\boldsymbol{X}^\top\boldsymbol{\beta})]\log\frac{1 - f^*(\boldsymbol{X}^\top\boldsymbol{\beta})}{1 - f^*(\boldsymbol{X}^\top\boldsymbol{\beta}')}\right\}$$

$$\leq \frac{1}{2}n \cdot \mathbb{E}_{\boldsymbol{X}}\left\{[1 + f^*(\boldsymbol{X}^\top\boldsymbol{\beta})]\left[\frac{1 + f^*(\boldsymbol{X}^\top\boldsymbol{\beta})}{1 + f^*(\boldsymbol{X}^\top\boldsymbol{\beta}')} - 1\right] + [1 - f^*(\boldsymbol{X}^\top\boldsymbol{\beta})]\left[\frac{1 - f^*(\boldsymbol{X}^\top\boldsymbol{\beta})}{1 - f^*(\boldsymbol{X}^\top\boldsymbol{\beta}')} - 1\right]\right\}.$$
(A.17)

In the last inequality, we utilize the fact that $\log z \leq z - 1$. Then by elementary calculation, we have

$$D_{KL}(P_{\boldsymbol{\beta}'}\|P_{\boldsymbol{\beta}}) \leq n \cdot \mathbb{E}_{\boldsymbol{X}}\left\{\frac{\big[f^*(\boldsymbol{X}^\top\boldsymbol{\beta}) - f^*(\boldsymbol{X}^\top\boldsymbol{\beta}')\big]^2}{\big[1 + f^*(\boldsymbol{X}^\top\boldsymbol{\beta}')\big]\cdot\big[1 - f^*(\boldsymbol{X}^\top\boldsymbol{\beta}')\big]}\right\}.$$
(A.18)

Using $|f(z)| \leq 1 - m$ and the Lipschitz continuity condition of $f$, we have

$$D_{KL}(P_{\boldsymbol{\beta}'}\|P_{\boldsymbol{\beta}}) \leq n \cdot \mathbb{E}_{\boldsymbol{X}}\left\{\frac{L^2\langle\boldsymbol{X}, \boldsymbol{\beta} - \boldsymbol{\beta}'\rangle^2}{m(1 - m)}\right\} = \frac{nL^2\|\boldsymbol{\beta} - \boldsymbol{\beta}'\|_2^2}{m(1 - m)} \leq \frac{2nL^2\xi^2}{m(1 - m)}.$$
(A.19)

Note that (A.17)-(A.19) hold for any $\boldsymbol{\beta}, \boldsymbol{\beta}' \in \overline{\mathcal{Q}}(p, s, \xi)$. We thus have

$$\max_{\boldsymbol{\beta}, \boldsymbol{\beta}' \in \overline{\mathcal{Q}}(p,s,\xi)} D_{KL}(P_{\boldsymbol{\beta}'}\|P_{\boldsymbol{\beta}}) \leq \frac{2nL^2\xi^2}{m(1 - m)}.$$

Now we proceed with (A.16) using the above result. The right hand side is thus lower bounded by

$$\frac{\sqrt{2}}{4}\xi\left(1 - \frac{2L^2n\xi^2/[m(1 - m)] + \log 2}{|\overline{\mathcal{Q}}(p, s, \xi)|}\right) \geq \frac{\sqrt{2}}{4}\xi\left(1 - \frac{2L^2n\xi^2/[m(1 - m)] + \log 2}{Cs\log(p/s)}\right),$$

where the last inequality is from $|\overline{\mathcal{Q}}(p, s, \xi)| \geq Cs\log(p/s)$. Finally, consider the case where the sample size $n$ is sufficiently large such that

$$n \geq \frac{m(1 - m)}{2L^2}\cdot\big[Cs\log(p/s)/2 - \log 2\big],$$

by choosing

$$\xi^2 = \frac{m(1 - m)}{2L^2n}\cdot\big[Cs\log(p/s)/2 - \log 2\big],$$
(A.20)

we thus have

$$\mathcal{R}(n, m, L, \mathcal{B}) \geq C'\cdot\frac{\sqrt{m(1 - m)}}{L}\cdot\sqrt{\frac{s\log(p/s)}{n}}$$

as required.