[Reviews · NeurIPS 2015]

Submitted by Assigned_Reviewer_1

This paper concerns the problem of estimating a vector \beta, from +/-1 measurements y_i which depend statistically on linear functions < x_i, \beta >, where the x_i are Gaussian random vectors. The statistical relationship between y_i and < x_i, \beta > is captured through a link function P[ y_i = 1 | X_i = x_i ] = 1/2 f( < x_i, \beta > ) + 1/2. This model is general enough to capture compressed sensing and phase retrieval problems with binary measurements. The paper assumes that f is completely unknown ahead of time, but that it satisfies certain moment conditions.

The paper shows how, under these moment conditions, to reduce the problem of estimating \beta to a sparse PCA problem, with covariance matrix generated by pairs of observations. The idea is to look at differences of x_i - x_{i'} and y_i - y_{i'}; the paper proves that the population covariance matrix of \delta_y \delta_x is a spiked identity matrix, where the spike is of the form \beta* \beta*^T.

This clever reduction appears to be the main contribution of the work. The paper then proposes and analyzes algorithms for solving the resulting sparse PCA problem. In the ``low-dimensional'' setting in which n > p, it analyzes the power method, while in the ``high-dimensional'' setting with \beta sparse and n < < p, it analyzes a combination of SDP relaxation and truncated power method. The paper presents bounds on estimation error and a minimax lower bounds. The rates of convergence are optimal, once the number of samples is large (n > s^2 log p).

In the high-dimensional regime, the proposed approach requires s^2 log p samples to recover an s-sparse vector. This matches the best known sample complexity for sparse phase retrieval (which may be optimal for efficient algorithms - nobody currently knows). However, it is suboptimal for one-bit compressed sensing.

This seems to be the main disadvantage of the proposed approach: for the sparse setting, there seems to be no way to use this reduction with less than s^2 samples; even though the problem might be solvable with about s samples by some other efficient algorithm. Because an improvement to the second stage of the algorithm would consist of a better algorithm for sparse PCA, this bottleneck seems insurmountable.

Because of the requirement that n > s^2 log p, I feel the title overclaims a bit. Of course, if there was some result indicating that about s^2 samples are necessary to solve problems in this class (maybe by reduction *from* sparse PCA?), this would go a long way to addressing this issue.

The paper obtains a minimax lower bound of the correct order \sqrt{s log p / n}. This lower bound requires a gap condition on the link function f, such that f is bounded away from 1. This restriction may only be technical in nature, but it seems to rule out most of the interesting examples. For example, in noisy one-bit compressed sensing, where y_i = sign( < x_i, w > + z_i ), with z_i independent gaussian noise, the link function f(t) approaches one as t -> \infty. [As the paper comments, the minimax lower bound also does not apply to the noiseless version of one bit compressed sensing]. It would be helpful if the authors' response can specify some interesting settings which do satisfy the gap condition.

There seems to be a minor notational issue in the theorems, as the algorithms can produce either an accurate estimate of \beta* or an accurate estimate of - \beta*. This ambiguity seems inevitable in this setting; please comment.

The two stage algorithm for the sparse regime involves SDP and then a nonconvex refinement. The SDP initialization limits its applicability to problems in several thousands of dimensions. This may be find for machine learning problems, but for phase retrieval (where the dimensionality is the number of pixels in the image to be recovered), it is limiting. Is it possible to avoid the SDP initialization, e.g., by simply solving one nonconvex problem?
Summary: The paper shows that under mild conditions, the problem of estimating a sparse \beta from +/-1 measurements y_i which depend statistically on \beta in an a-priori unknown manner, can be reduced to a sparse PCA problem. The reduction is clever, and can be used to produce estimators with guaranteed good performance, although the number of samples required is bottlenecked by the sample complexity of the sparse PCA problem.

Submitted by Assigned_Reviewer_2

The paper studies the problem of estimating parameters in a linear model, where the observations have undergone a nonlinear transform, are noisy and have been quantized to a single bit. It unifies logistic regression, flipped logistic regression, one-bit compressed sensing and one-bit phase retrieval in the sense that that no assumptions are made on the nonlinear transform (except a mild restriction on its moments).

The paper is well written and presents an interesting, easy and attractive solution for the problem at hand. The paper has in general strong theoretical arguments. The model claims to be more generic than the one used in logistic regression, one-bit compressed sensing and one-bit phase retrieval. I agree on this claim, however, in (for example) a phase retrieval problem, wouldn't it be more efficient to use the explicit link function provided for that specific problem, than letting the link function be unknown as you do in your model?

Since the paper is restricted to single-bit problem, I think this should be reflected in the title. As it is now, I think the title is too general. Further, how restrictive is your assumption that the observation is quantized to a single bit? I would expect that for example Lemma 3.2 would hold also if y is not quantized, but maybe I am wrong. Is it possible to give an intuition why the quantization is important for your results or when it could be relaxed?

I also try to understand what your assumption on the link function means. Does phi(f) \neq 0 have any practical meaning? If no assumption is made on the link function I would expect you to overfit on the training data. Could you possible comment on this?

The weak point of the paper is the numerical section which is fairly short. I would have liked to see a comparison with logistic regression, one-bit compressed sensing and/or one-it phase retrieval on a certain classification task. With the present results I am still not convinced why I should use your model instead of one of the models that you claim to unify.

Minors: - Row 23: "...sparse ,and..." -> "...sparse, and..."

- Row 98/99 "primarily in interested in" -> "primarily interested in"
Summary: The paper is well written with its numerical section as the weak point. Nevertheless, I recommend to accept this paper.

Submitted by Assigned_Reviewer_3

Some suggestions for improvement: (i) Consider including a high level sketch of the proofs of Theorems 3.5-3.7 in the main body of the paper. (ii) Consider including a wider suite of numerical experiments (including comparisons with other methods such as [30].
Summary: The paper proposes a class of algorithms, based on a novel spectral estimation procedure, for estimating high-dimensional signals under the single-index observation model. The paper is well-written, the techniques are sound, and the contributions are clear and compelling.

Submitted by Assigned_Reviewer_4

The paper discusses a spectral-based approach that achieves optimal computational and statistical rates of convergence for estimating the coefficients in a specific class of single index model encompassing common examples such as logistic regression, one-bit compressed sensing and phase retrieval. The results are separated according to a low-dimensional setting (n > p) and a high-dimensional, sparse setting which is considerably more involved. A minimax lower bound is derived that establishes optimality of the upper bounds. Simulated data are used to verify the rates of convergence.

Quality/Clarity: The technical quality of the paper is high. The presentation is excellent as the reader is guided well through the results. In particular, the many references cited by the authors make it easy to relate their results to earlier work.

Originality/Significance: The spectral-based estimation approach in this context appears quite original to me even though the general idea may have been borrowed from earlier work. Compared to the closely related work of Plan \& Vershynin, the approach in the present paper applies to a broader class of single index models.

Moreover, the authors unify computational and statistical properties. On the negative side, I feel that the authors have not made sufficient effort to show the practical benefits of their approach. The experimental results do not contain comparisons to state-of-the art methods.

Specific comments/questions: - it should be stressed more that as distinguished from classical

approaches to single index models, the authors' approach is limited to inference of the coefficients. Since the link function is not estimated, prediction of future observations is not adressed.

- in Section 3.2 the authors discuss two estimators for $\beta*$ depending on the sign of the functional $\phi(f)$. Since the link function $f$ is unknown, how does one decide for one of the estimators in practice ? - The formatting of the Algorithms and the text of Section 3.3 should be improved

Summary: Technically strong paper with valuable results.

Author Feedback
Author rebuttal: We are very grateful to the careful and throughout reviews from all reviewers.

Reviewer_1
For the lower bound and the gap condition: flipped logistic regression (and its high-dim analog (Sec 2)) where signs are flipped with a certain prob. p_e > 0, satisfies the gap condition, and in fact many such robust examples (including flipped one-bit CS) do. More generally, for the bounding assumption about f in our lower bound, it's worth noting that the established lower bound \sqrt{s\logp/n} never holds if we completely remove this constraint since the best algorithm for noiseless CS has better convergence rate.

Regarding sample complexity vs convergence rate: as the reviewer points out, our algorithm has sample complexity s^2*log p which for some problems we know is not optimal; the convergence rate, however, is \sqrt{s*logp/n}, which is optimal as our minimax lower bound demonstrates. It is entirely possible that s^2*log p is a lower bound of sample complexity for polynomial time algorithm as no algorithm is known to work for sparse phase retrieval, which is easier than the general problem we consider, with sample s*log p. We believe proving (or disproving) this lower bound would be very interesting.

The sign ambiguity in the recovery is indeed unavoidable as we do not know link function. We will make this more explicit in the paper.

The global convergence of the truncated power method requires a good initialization theoretically and empirically. It is unknown how to use a one-phase non-convex optimization to solve our problem. By lifting dimensions, the memory requirement becomes a serious issue for large scale problems. It's unknown how to avoid it at this point. For running time, it's worth noting that we can stop early when solving the SDP because we only need a constant approximation and obtaining the global optimum of the SDP is not necessary. And also, as we pointed out, the SDP can be easily solved by using ADMM.

Reviewer_2
It's possible that if the link function is known a priori, then a specialized algorithm taking advantage of this knowledge might have a more efficient solution than our general algorithm. For instance, to solve sparse logistic regression, one can choose to solve a regularized convex optimization problem based on ML. Our aim is to address the scenario where the link function is unknown or the model is dirty (which is not uncommon in practice).

By restricting the response to single bit, the model can be fully encoded in a single link function f. Then the sufficient recovery conditions can be established elegantly in terms of f's properties. For arbitrary response y, one may assume y = f( < x,\beta > + e) where f is a nonlinear function and e is noise with certain distribution. Then the recovery conditions need to be discussed on unknown function and distribution, which could be tedious and complicated in the current stage.

For odd functions f, the success condition's meaning is easy to explain intuitively: phi(f) \ne 0 says f is not a constant (no algorithm would work). Regarding over-fitting: as remarked above, we do pay a price for not knowing the link function (we need more samples). But the mildness of the price is what we see as the main importance of the results and hence contribution. The reviewer's point leads to an interesting open problem, which is to characterize the necessary conditions for any possible algorithm.

The reviewer makes a good point on testing some models in classification task, and we plan on including this in the appendix. We would like to mention that most models we consider such as one bit compressive sensing have received significant attention in ML, signal processing and applied math communities.

Reviewer_3
Regarding computational results: we agree that more are needed to assess the practical impact. At this stage the theory was the central focus, but it is now something we working on. We do not estimate the link function, and hence the goal is inference, not prediction. We will make this more clear.

For different sign of phi(f), we have two different spectral formulations as we pointed out in our paper. In practice, one can tell which one is right by inspecting the spectrum of both matrices. The right one should have an obvious gap (if the number of samples is sufficiently large) between the first and second eigenvalues.

Reviewer_4
It would be very nice to remove the Gaussian assumption, though it seems difficult given the broad generality of settings to which our model applies. Also, it's worth noting that as pointed in Plan and Vershynin's paper [30], extending the Gaussian assumption to subGaussian is very difficult even for the special problem of one-bit compressive sensing. As we've mentioned above, we are working on expanding the computational section.

Reviewer_5 & Reviewer_6
Both reviewers provide positive feedback to our work. Their suggestions for improvement will be considered for our final paper.